# Attenuated Influenza Virions Expressing the SARS-CoV-2 Receptor-Binding Domain Induce Neutralizing Antibodies in Mice

**DOI:** 10.3390/v12090987

**Published:** 2020-09-05

**Authors:** Andrea N. Loes, Lauren E. Gentles, Allison J. Greaney, Katharine H. D. Crawford, Jesse D. Bloom

**Affiliations:** 1Division of Basic Sciences and Computational Biology Program, Fred Hutchinson Cancer Research Center, Seattle, WA 98109, USA; aloes@fredhutch.org (A.N.L.); lgentles@uw.edu (L.E.G.); agreaney@uw.edu (A.J.G.); kdusenbu@fredhutch.org (K.H.D.C.); 2Howard Hughes Medical Institute, Seattle, WA 98103, USA; 3Department of Microbiology, University of Washington, Seattle, WA 98195-7735, USA; 4Department of Genome Sciences, University of Washington, Seattle, WA 98195, USA; 5Medical Scientist Training Program, University of Washington, Seattle, WA 98195, USA

**Keywords:** SARS-CoV-2, influenza, intranasal, live attenuated vaccine, spike, RBD

## Abstract

An effective vaccine is essential for controlling the spread of the SARS-CoV-2 virus. Here, we describe an influenza virus-based vaccine for SARS-CoV-2. We incorporated a membrane-anchored form of the SARS-CoV-2 spike receptor binding domain (RBD) in place of the neuraminidase (NA) coding sequence in an influenza virus also possessing a mutation that reduces the affinity of hemagglutinin for its sialic acid receptor. The resulting ΔNA(RBD)-Flu virus can be generated by reverse genetics and grown to high titers in cell culture. A single-dose intranasal inoculation of mice with ΔNA(RBD)-Flu elicits serum neutralizing antibody titers against SAR-CoV-2 comparable to those observed in humans following natural infection (~1:200). Furthermore, ΔNA(RBD)-Flu itself causes no apparent disease in mice. It might be possible to produce a vaccine similar to ΔNA(RBD)-Flu at scale by leveraging existing platforms for the production of influenza vaccines.

## 1. Introduction

The rapid spread and severity of disease caused by the SAR-CoV-2 virus highlights the need for effective control measures [1,2,3]. A safe, effective, and scalable vaccine is the most promising way to limit the public health impact of SARS-CoV-2.

The influenza virus is a promising platform for vaccine development. Attenuated influenza viruses are already used as vaccines against influenza itself [4,5,6,7]. In addition, well-established influenza reverse genetics systems make it easy to incorporate foreign genes into the viral genome [8,9,10,11,12]. In preclinical animal models, influenza virions expressing foreign antigens induce antibodies against the causative agents of a variety of diseases, including the West Nile virus, *Bacillus anthracis,* botulinum neurotoxin, and HIV [13,14,15,16]. Intranasal infection with influenza viruses induces mucosal immune responses not only within the respiratory tract, but also in mucosal surfaces distal from the site of immunization [15], which may prove important for SARS-CoV-2 [17]. Finally, there is already existing infrastructure for the large-scale production of influenza virions for use in vaccines.

The receptor binding domain (RBD) of the spike glycoprotein is a key antigen candidate for SARS-CoV-2 vaccines [18,19,20,21,22,23]. Many of the most potent neutralizing antibodies against SARS-CoV-2 target the RBD [19,24,25,26,27]. Additionally, RBD is small in size (~200 aa), folds autonomously, and is therefore an attractive candidate for protein subunit-based approaches to SARS-CoV-2 vaccinations [19].

Here, we have incorporated a membrane-anchored form of the SARS-CoV-2 spike RBD into the genome of an attenuated influenza A virus. Cells infected with the resulting virus express high levels of RBD on their surface. The inoculation of mice with the virus induces substantial titers of neutralizing antibodies (~1:200) without causing disease. This influenza virus-based vaccine platform could be useful for combating SARS-CoV-2.

## 2. Materials and Methods

### 2.1. Plasmids

The ∆NA(RBD) construct, described in Figure 1A and the first results subsection below, used the protein sequence for the receptor binding domain (RBD) of spike from SARS-CoV-2 (isolate Wuhan-Hu-1, Genbank accession number MN908947, residues 331-531). The nucleotide sequence was codon-optimized as previously described [28]. To facilitate the surface expression of the RBD, the murine immunoglobulin H chain V-region leader sequence was added to the N-terminus. The transmembrane region and cytoplasmic domain of murine B7.1 (CD80) was added to the C-terminus to tether this protein to the cell membrane [29]. To incorporate the sequence for this membrane-anchored RBD into an influenza viral RNA (vRNA), its coding sequence was flanked by the coding and noncoding regions of the packaging signal from the A/WSN/33 influenza NA segment (113 and 126 bp at the 3′ and 5′ ends of the vRNA, respectively) [30,31]. Several mutations were introduced into the 3′ and 5′ packaging signals, including the ablation of ATG start codons in the 3′ vRNA, the ablation of a putative cryptic bacterial translation initiation site, and the introduction of cloning restriction sites between the 3′ and 5′ packaging signals and coding sequences. This construct was cloned into a pHH vector and expressed via the Pol I promotor [9] to generate the viral stock used for murine infections. Subsequent reverse genetics experiments were performed with this gene cloned into the pHW2000 backbone [32] (pHW_NAflank_RBD_B7-1). A Genbank file for pHW_NAflank_RBD_B7-1 is provided in (Appendix A).

The ∆NA(GFP) segment, containing green fluorescent protein (GFP) in place of neuraminidase (NA), was based on the FLU-NA-GFP described in Rimmelzwaan and others [31], except that the NA sequences come from A/WSN/33 rather than A/PR/8/34. GFP was similarly flanked by the noncoding regions and terminal coding nucleotides from the A/WSN/33 influenza NA segment. The 3′ vRNA sequence contained three nucleotide differences and one nucleotide deletion relative to the flanking RBD in the pHW_NAflank_RBD_B7-1 construct. The sequence for pHH_NAflank_eGFP is available in the supplementary information (Appendix A).

Additional viral genes from A/WSN/33 were expressed from bidirectional reverse genetics plasmids [9], kindly provided by Robert Webster of St. Jude Children’s Research Hospital. The H3 HA protein containing a Y98F mutation from A/Aichi/2/1968 was also cloned into the bidirectional pHW2000 backbone, and additional mutations G379W (Genbank sequence is in Appendix A) or R453G, were introduced into this construct to improve the rescue of neuraminidase-deficient viruses. The HA mutations are named in the canonical H3 ectodomain numbering scheme.

For cell surface expression studies, the protein sequence for the SARS-CoV-2 spike from (isolate Wuhan-1, Genbank NC_045512) was codon-optimized and contained a cytoplasmic tail truncation that removes the last 21 amino acids, as described previously [33]. Spike and the protein sequence for the membrane-anchored RBD in ∆NA(RBD) were each cloned into an expression plasmid (HDM) which places the gene under the control of a cytomegalovirus (CMV) promoter, to generate plasmids HDM-SARS2-Spike-delta21 [33] and HDM_Spike_RBD_B7-1 (Appendix A).

### 2.2. Generation of Viruses

Influenza viruses were generated by reverse genetics, as previously described [34]. Briefly, we transfected a coculture of 293T cells and MDCK-SIAT1-TMPRSS2 cells [34] with reverse genetics plasmids encoding the internal segments from the A/WSN/33 virus (PB1, PB2, PA, NP, M, NS), the hemagglutinin (HA) segment from the A/Aichi/2/1968 virus with an amino acid mutation in the receptor binding site (Y98F), and either the ∆NA(RBD) or ∆NA(GFP) segment described above, or the NA segment from the A/WSN/33 virus. Cells were plated at a density of 5 × 10^5^ 293T cells, and 0.5 × 10^5^ MDCK-SIAT1-TMPRSS2 cells per well in a 6-well dish in D10 (Dulbecco modified Eagle medium supplemented with 10% heat-inactivated fetal bovine serum (FBS), 2 mM L-glutamine, 100 U of penicillin/mL, and 100 µg of streptomycin/mL). The next day, 200 ng of each plasmid was transfected into the cells using the BioT transfection reagent (Bioland Sci, Paramount, CA, B01-02). Then, 12–18 h post-transfection (hpt), the medium was changed to influenza growth media (IGM; Opti-MEM supplemented with 0.01% heat-inactivated FBS, 0.3% bovine serum albumin (BSA), 100 U of penicillin/mL, 100 µg of streptomycin/mL, and 100 µg of calcium chloride/mL). Viral supernatants were collected approximately 72 hpt and the titer was determined by a fifty-percent tissue-culture infective dose (TCID50).

To generate inoculum for murine infections, viruses were expanded in low serum media. We infected MDCK-SIAT1-TMPRSS-2 cells at a low multiplicity of infection (MOI = 0.02 TCID50). The medium was changed to OptiMEM at 24 h postinfection (hpi) and the virus was collected 48–72 hpi. Viral supernatants were clarified to remove cells by filtering through a 0.45 µM syringe filter, aliquoted, and stored at −80 °C. Viruses were titered by TCID50 prior to use in mouse experiments. The ∆NA(RBD)-Flu virus reached a titer of 1.7 × 10^4^ TCID50/uL following an expansion in OptiMEM. To verify that ∆NA(RBD) was stably maintained in NA segment, RNA was extracted from the viral supernatant with RNeasy kit (Qiagen, Germantown, MD, 74104), and reverse transcribed to produce cDNA (Agilent, Santa Clara, CA, 200820). The NA segment was amplified by PCR and visualized with a 1% agarose gel.

### 2.3. Flow Cytometry

293T cells were seeded at 6 x 10^5^ cells/well in a 6-well plate, and cells were transfected twenty-four hours later with 500 ng of either HDM-SARS-Spike-delta21 or HDM_Spike_RBD_B7-1 plasmids and 1.5 ng of Transfection Carrier DNA (Promega, Madison, WI, E4881) using BioT reagent, according to manufacturer’s protocol. Alternatively, the media was changed to IGM and cells were infected with either ∆NA(RBD)-Flu or ∆NA(GFP)-Flu. At 18 hpi/hpt, the media was removed from cells, and cells were washed with phosphate buffered saline (PBS), dissociated from the plate with enzyme-free dissociation buffer (Thermo Fisher, Waltham, MA, 13151014), harvested by centrifugation at 1200 × *g* for 3 min, and washed in FACS buffer (PBS + 1%, Bovine Serum Albumin). Cells were stained with either recombinant biotinylated ACE2 ectodomain (ACROBiosystems, Newark, DE, AC2-H82E6) or CR3022 antibody (kindly provided by Neil King and Mike Murphy, University of Washington, Institute for Protein Design) for 1 h at room temperature, washed with FACS buffer, resuspended in secondary stain, a 1:200 dilution of PE-conjugated streptavidin (Thermo Fisher, S866) or PE-conjugated Goat Anti-Human IgG (Jackson ImmunoResearch Labs, West Grove, PA, 109-115-098), and incubated on ice for 1 h. Cells were then washed twice in FACS buffer. Cells infected with the replicative virus were fixed in 1% paraformaldehyde in FACS buffer for 15 min on ice and washed twice more prior to FACS analysis. The RBD surface expression was assessed by measuring PE-positive cell populations with a BD FACSCanto II instrument. For each sample, 10,000 events were collected, and the data shown were gated to select singleton events. The analysis and compensation were performed using FlowJo v10.7. The histograms shown are for a single well of stained cells. However, the verification of the surface expression of RBD upon infection with ΔNA(RBD)-Flu was replicated on a separate day.

### 2.4. Animal Studies

Seven week old female BALB/cJ mice (Jackson Labs, Sacramento, CA) were anesthetized intraperitonially with 100 mg/kg of ketamine + 10mg/kg of xylazine in PBS and infected intranasally with 50 uL of either a high (8 × 10^5^ TCID50) or low (8 × 10^4^ TCID50) dose of ΔNA(RBD)-Flu, or with ΔNA(GFP)-Flu virus (8 × 10^4^ TCID50), or mock infected with OptiMEM. Four animals were inoculated per condition. At 14 and 21 days postinfection, blood was collected from mice by a retro-orbital bleed and/or cardiac puncture at the terminal time point. The blood samples clotted at room temperature for 30–60 min and then were centrifuged at 1000× *g* for 10 min at 4 °C to separate the cellular debris and plasminogen from the sera. The sera were stored at −80 °C until use. Animal work was conducted according to protocol PROTO201900016 (1893), approved by Fred Hutchinson Cancer Research Center Institutional Animal Care and Use Committee (9 June 2020).

### 2.5. Spike Pseudotyped Lentivirus Neutralization Assays

SARS-CoV-2 spike-pseudotyped lentivirus neutralization assays were performed as previously described [33,35], with slight modifications. The spike protein of SARS-CoV-2 that was used to produce pseudotyped lentivirus contained a cytoplasmic tail truncation that removes the last 21 amino acids [33]. Infections of 293T-ACE2 cells were performed in poly-L-lysine coated plates. The sera were heat-inactivated for 30 min at 56 °C immediately prior to use. Three-fold dilutions of sera were done starting at a 1:20 dilution. The serum was incubated with the virus for 1 h at this concentration (which was used to calculate IC50s) and then diluted 2:3 when transferred onto cells. The luciferase activity was measured at 65 hpi following a transfer to opaque black-bottom plates. Samples were run in duplicate and each plate included no-serum and no-virus controls. The fraction infectivity was calculated by normalizing the luciferase reading for each sample by the average of two no-serum control wells in the same row. All neutralization curves show the mean and standard error of duplicate curves run on the same 96-well plate. The neutralization curves were plotted using the neutcurve Python package (https://jbloomlab.github.io/neutcurve/, 0.3.1), which fits a two-parameter Hill curve, with the top baseline fixed to one and the bottom baseline fixed to zero.

### 2.6. Influenza Neutralization Assays

Sera were treated with receptor-destroying enzyme (RDE) to ensure that the virus would not bind to residual sialic acids present in the serum. One vial of lyophilized RDE II (Denka Seiken, Tokyo, JP, 370013) was first resuspended in 20 mL PBS. We then incubated 10 μL of serum with 30 μL of RDE solution at 37 °C for 3 h, heat-inactivated the serum and RDE by incubating at 56 °C for 30 min, centrifuged the serum at 6000× *g* for 2 min to pellet any precipitated material and collected the supernatant. Neutralization was measured against the virus containing H3 from the A/Aichi/2/1968 virus with two mutations, Y98F and G379W, carrying GFP in the PB1 segment, and all other internal genes from A/WSN/33. The neutralization assays were performed in MDCK-SIAT1-CMV-PB1 cells using this GFP-expressing virus as described previously [36,37,38,39]. As with spike pseudotyped lentivirus neutralization assays, the curves were plotted and IC50s were calculated using the neutcurve Python package.

## 3. Results

### 3.1. Engineering and Characterization of Influenza Virus Encoding SARS-CoV-2 RBD

We set out to incorporate a membrane-anchored form of the receptor binding domain (RBD) of SARS-CoV-2 spike into the influenza genome. To enable the cell-surface display of the RBD protein, we appended a mammalian signal peptide to the N-terminus of the RBD and a transmembrane region and cytoplasmic domain from murine B7.1 to the C-terminus of the RBD. The murine B7.1 sequence was used as it has previously been shown to induce high levels of surface expression of chimeric proteins [29]. We engineered the neuraminidase (NA) segment of the A/WSN/33 influenza virus to replace the NA coding sequence with a sequence encoding the membrane-anchored RBD and named the resulting construct ΔNA(RBD) (Figure 1A). To facilitate the packaging of this segment into influenza virions, the packaging signals located at the NA segment termini, which the span coding and noncoding regions of vRNA, were retained (Figure 1A). These terminal packaging signals comprised 113 and 126 bp at the 3′ and 5′ ends of the vRNA, respectively [30,31]. Mutations were introduced in these regions to remove the start codons upstream of the RBD coding sequence, as well as eliminate a cryptic bacterial promoter (see the plasmid map in Appendix A). Restriction sites were inserted immediately adjacent to the NA packaging signal on both the 5′ and the 3′ ends of the coding sequence.

We used reverse genetics to generate influenza virions encoding the ΔNA(RBD) segment and named the resulting virus ΔNA(RBD)-Flu. In addition to the ΔNA(RBD) segment, these viruses contained the hemagglutinin (HA) segment from the A/Aichi/2/1968 (H3N2) strain with an amino acid mutation (Y98F) that reduces the affinity for HA’s sialic acid receptor [40]. The Y98F mutation was included for two reasons: first, we anticipated that it would improve the growth of viruses that lacked an NA, which typically counterbalances HA’s receptor-binding activity [36,40]; second, because viruses with this mutation are known to be severely attenuated in mice [4]. The remaining influenza genes of ΔNA(RBD)-Flu were derived from the mouse- and lab-adapted strain A/WSN/33 (H1N1). The viral supernatant from reverse genetics was blind passaged twice in MDCK-SIAT1-TMPRSS2 cells at three-day intervals. Upon the final passage, the cytopathic effect was evident in the well containing ΔNA(RBD)-Flu, suggesting the presence of a replicating virus.

Next, we verified that the RBD sequence was stably maintained in the NA segment. We extracted RNA from the viral supernatant from the final passage, and reverse transcribed and PCR-amplified the HA and NA segments (Figure 1B). The size of the NA segment isolated from ΔNA(RBD)-Flu was consistent with the expected size of this segment. Sanger sequencing was performed on both the NA and HA segments to identify any potential tissue-culture adaptations. No mutations were identified in the ΔNA(RBD) segment. However, a single amino acid substitution was present in HA (G379W), which we speculated might aid the growth of these engineered virions. We validated that this mutation, and an additional HA mutation isolated in a virus with no NA segment (R453G), improved the titers of ΔNA(RBD)-Flu generated by reverse genetics, although the titers were still lower than for a virus with the same HA and an intact NA segment (Figure 1C). The mechanism for how these mutations in HA2 might improve titers from reverse genetics is not known. For all subsequent experiments, the originally isolated and amplified ΔNA(RBD)-Flu, containing H3_Y98F/G379W, was used. As a control, we used a virus containing this HA (H3_Y98F/G379W) and GFP in the NA segment (ΔNA(GFP)-Flu).

### 3.2. Cells Infected with ΔNA(RBD)-Flu Express High Levels of RBD on Their Surface

We next set out to determine if cells transfected with the membrane-anchored RBD gene or infected with ΔNA(RBD)-Flu expressed RBD on their surface. To do this, we transfected cells with a mammalian protein expression plasmid encoding the membrane-anchored RBD or the SARS-CoV-2 spike (lacking the last 21 amino acids of the cytoplasmic tail). We also infected cells with either ΔNA(RBD)-Flu or ΔNA(GFP)-Flu. We stained the cells with a soluble recombinant version of ACE2, the host receptor that is bound by the SARS-CoV-2 spike’s RBD to mediate viral entry. We separately stained the cells with the antibody CR3022, which binds to the RBD in a region which does not overlap with the ACE2 binding site [41,42]. There were high levels of RBD expression on the surface of transfected and infected cells labeled with both ACE2 and CR3022 (Figure 2). Notably, cells transfected with the membrane-anchored RBD had substantially higher staining of RBD on their surface than cells transfected with spike, and cells infected with ΔNA(RBD)-Flu had even higher levels of RBD than the transfected cells (Figure 2).

### 3.3. ΔNA(RBD)-Flu Induces Neutralizing Antibodies Against SARS-CoV-2 and Influenza Virus Without Causing Disease in Mice

To determine if the ΔNA(RBD)-Flu virus induces an anti-RBD antibody response in vivo, we intranasally infected four groups of four mice with either a high dose (8 × 10^5^ TCID50) or lower dose of ΔNA(RBD)-Flu (8 × 10^4^ TCID50), a control virus expressing GFP in the NA segment (ΔNA(GFP)-Flu, at 8 × 10^4^ TCID50), or performed a mock infection (Figure 3A). Mice were monitored for seven days postinfection for signs of illness, including reduced activity, ruffled fur, hunched posture, and weight loss (Figure 3B). No signs of disease or weight loss were observed in any group (Figure 3B).

We collected sera from the mice at 14 and 21 days postinfection (dpi) and measured anti-SARS-CoV-2 titers using a spike-pseudotyped lentivirus neutralization assay [33,35]. Neutralizing responses against SARS-Cov-2 were detected in mice infected with the ΔNA(RBD)-Flu at both 14 and 21 dpi (Figure 3C, curves available in Appendix A). Three of the four mice in each dose group had neutralizing titers at 14 dpi, while all mice infected with ΔNA(RBD)-Flu show neutralizing titers by 21 dpi (Figure 3C). The neutralizing titers were higher for mice inoculated with a larger dose of ΔNA(RBD)-Flu, although the difference was not statistically significant. As expected, no neutralizing antibodies against SARS-CoV-2 Spike were detected in either the mock or ΔNA(GFP)-Flu infected mice (Figure 3C). These data indicate that a single intranasal inoculation with ΔNA(RBD)-Flu induces a rapid production of neutralizing antibodies against SARS-CoV-2.

Finally, we tested if the mice had also mounted an immune response to influenza virus. We performed influenza virus neutralization assays with blood collected at 21 dpi, using an influenza virus with the H3 from A/Aichi/2/1968 with Y98F and G379W mutations (the same mutations present in the HA ΔNA(RBD)-Flu). As expected, high levels of neutralizing antibodies against influenza virus were present in mice infected with either ΔNA(RBD)-Flu or ΔNA(GFP)-Flu (Figure 3D).

## 4. Discussion

Here we describe an engineered influenza virus, ΔNA(RBD)-Flu, that expresses a membrane-anchored form of SARS-CoV-2 Spike RBD from its NA segment. The ΔNA(RBD)-Flu virus stably maintains the gene encoding the RBD over multiple passages, and cells infected with this virus express high levels of RBD on their surface. An intranasal inoculation of mice with a single dose of ΔNA(RBD)-Flu elicits neutralizing antibodies against SARS-CoV-2 in mice, with titers comparable to those observed in humans following natural infection [33,43,44] or an administration of two doses of a mRNA-based vaccine in clinical trials [20]. Notably, the ΔNA(RBD)-Flu elicited these neutralizing antibody titers after just a single intranasal inoculation at a viral dose that is lower than that typically used for live-attenuated influenza vaccines [4,13,45]. It therefore seems possible that neutralizing antibody titers against SARS-CoV-2 could be further enhanced by using a higher viral dose in a single inoculation, and/or by boosting either with recombinant RBD or a second administration of a ΔNA(RBD)-Flu variant containing a different HA protein.

Our study also has several important limitations. First, we examined the effect of ΔNA(RBD)-Flu vaccination only in influenza-naïve mice. For other live attenuated vaccines, a pre-existing immunity against the vaccine vector (influenza virus, in this case) can reduce effectiveness [46,47]. This limitation could potentially be overcome by creating variants of ΔNA(RBD)-Flu that contain HA subtypes to which the human population is naïve, although we did not attempt that here. Second, we performed experiments using only a single strain of mice (BALB/cJ) and have not examined the results of vaccination with ΔNA(RBD)-Flu in nonhuman primates or other animal models. Third, although the ΔNA(RBD)-Flu grew to reasonably high titers that were sufficient for our experiments, the titers were still lower than those obtained using viruses with an intact NA segment. Fourth, although we demonstrated high SARS-CoV-2 neutralizing antibody titers at three weeks postvaccination, we did not perform experiments to examine the durability of these titers over longer timeframes. Finally, while the presence of neutralizing titers against SARS-CoV-2 is correlated with protection against infection in animal models [48,49] and preliminarily in human studies [50], we did not directly test if the neutralizing antibodies elicited by ΔNA(RBD)-Flu are protective against viral challenge.

Despite these limitations, vaccines based on ΔNA(RBD)-Flu have several possible advantages. First, the vaccine induces neutralizing antibodies with a just single dose. Second, live influenza viruses induce mucosal immune responses [10,13,15,51] that may be important for protective immunity to SAR-CoV-2 [17,52]. Third, it might be possible to leverage additional influenza virus-engineering approaches such as those described in [11] to generate virions that express NA as well as RBD, and could serve as dual influenza- and SARS-CoV-2 vaccines. Finally, a major challenge to the rapid development and deployment of a vaccine for COVID-19 is scaling production in order to distribute a large number of doses worldwide. There is already existing infrastructure for the large-scale production of influenza virus-based vaccines which could in principle be extended to also produce live virus vaccines similar to ΔNA(RBD)-Flu at scale [23].

## 5. Patents

A.N.L., L.E.G., A.J.G., and J.D.B are listed as inventors on a provisional patent application based on the studies presented in this paper.

## Figures and Tables

**Figure 1 viruses-12-00987-f001:**
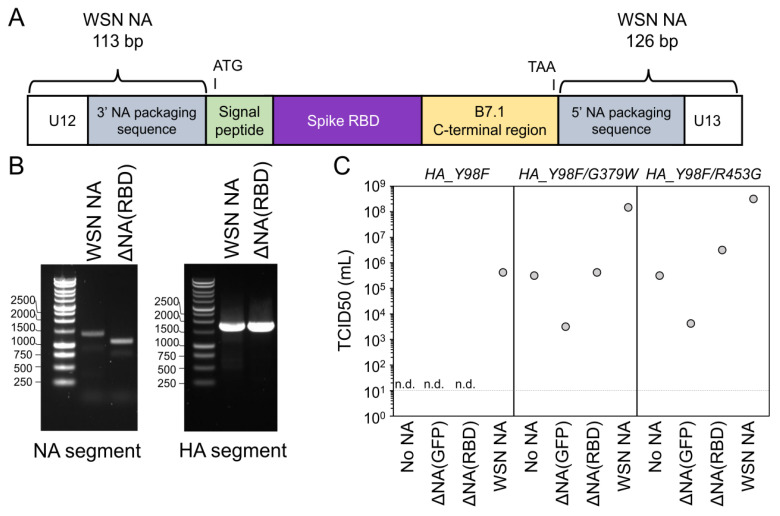
Design and characterization of a neuraminidase-deficient influenza virus containing receptor binding domain (RBD) of SARS-CoV-2 spike protein. (**A**) Schematic diagram of construct encoding SARS-CoV-2 Spike RBD with a eukaryotic signal peptide and the transmembrane region and cytoplasmic tail from murine B7.1 [29] in the neuraminidase (NA) segment of influenza virus. Sequence for RBD is flanked at the 5′ and 3′ ends by the noncoding and coding terminal packaging regions of the NA viral RNA (vRNA) segment from A/WSN/33. Start (ATG) and stop codons (TAA) are shown to indicate the region that is expressed in cells. Note that the schematic is drawn in the orientation of the positive-sense mRNA, not the negative sense vRNA. (**B**) Agarose gel of amplified NA and hemagglutinin (HA) segments from ΔNA(RBD)-Flu. For comparison, HA and NA segments from virus containing a wildtype NA segment are also shown. The expected sizes of bands are: wildtype NA = 1409 bp, ΔNA(RBD) = 1125 bp, and HA = 1765 bp. (**C**) The effect of introducing G379W and R453G into H3_Y98F HA on viral titers. TCID50/mL is shown for reverse genetics for viruses with no NA segment, or the indicated NA segment. Values shown represent titers from a single reverse genetics experiment. Limit of detection is indicated with a horizontal dashed line. Samples where no cytopathic effect was observed are indicated with “n.d.” (none detected).

**Figure 2 viruses-12-00987-f002:**
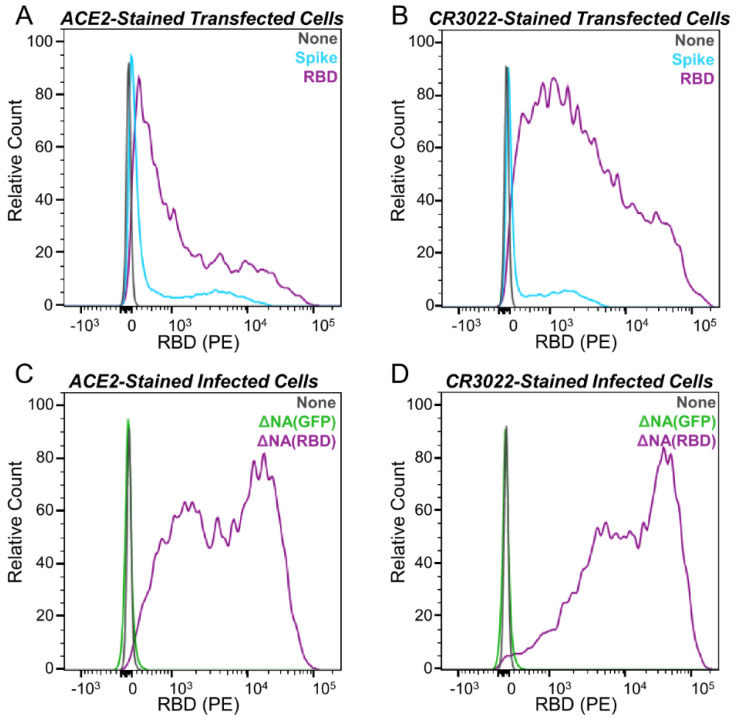
Infection with ΔNA(RBD)-Flu leads to expression of RBD on the surface of 293T cells. Histograms, normalized to mode, of flow cytometry measurements showing RBD expression on the cell surface for: (**A**) Cells transfected with the mammalian expression plasmids HDM-SARS-Spike-delta21 or HDM_Spike_RBD_B7-1 and stained with ACE2 or (**B**) stained with CR3022 antibody, (**C**) Cells infected with ΔNA(RBD)-Flu or ΔNA(GFP)-Flu and stained with ACE2 or (**D**) CR3022 antibody. Data shown are from a single population of stained cells.

**Figure 3 viruses-12-00987-f003:**
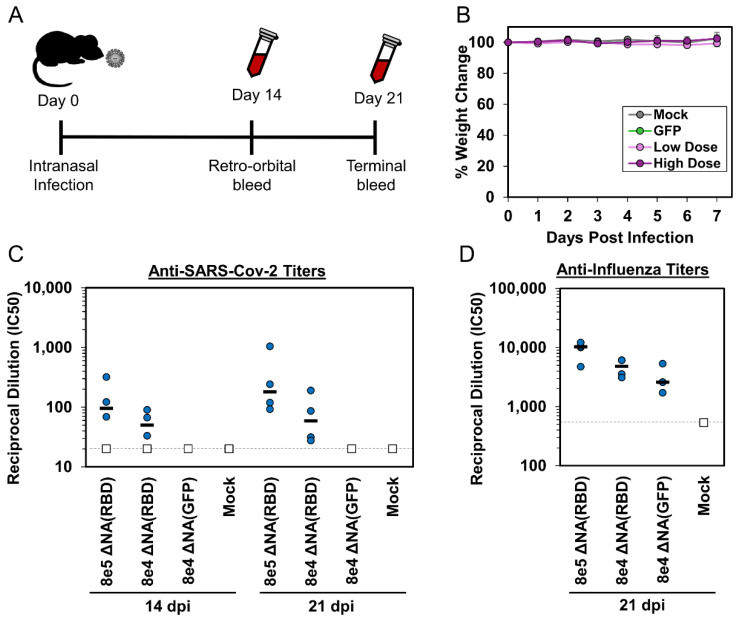
The ΔNA(GFP)-Flu virus induces neutralizing antibodies against influenza and SARS-CoV-2. (**A**) Schematic of timeline for intranasal infection and blood collection from mice. (**B**) Weight of individual mice was measured for the first seven days postintranasal infection to monitor disease severity for any signs of disease. Values shown are percent weight change from initial weight prior to infection on Day 0. Points are average and error bars are standard deviation of each group. Titers of neutralizing antibodies against (**C**) SARS-CoV-2 Spike-pseudotyped lentiviral particles at 14 and 21 days postinfection, and (**D**) influenza virus with the H3 HA from A/Aichi/2/1968 at 21 days postinfection. Y-axis values show reciprocal dilution of the fifty-percent inhibitory concentration (IC50). Blue circles denote titer from individual mice in each group, horizontal line indicates median titer for the group. Samples were run in duplicate. The dashed line indicates the limit of detection, points at that limit (white squares) are the lower bound. There are four mice in each group.

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
