# Peer review of "Attenuated Influenza Virions Expressing the SARS-CoV-2 Receptor-Binding Domain Induce Neutralizing Antibodies in Mice"

_viruses, 2020, doi:10.3390/v12090987_

Round 1

Reviewer 1 Report

In their manuscript, Loes and colleagues describe the capacity of engineered influenza viruses to elicit the production of neutralising, RBD-targeting SARS-CoV-2 antibodies in a murine model of influenza virus infection.

After presenting their construct and the role of mutations in NA for the in vitro propagation of virus, the authors describe how this engineered virus can induce the expression of the SARS-CoV-2 RBD at the surface of infected cells. Specifically, they demonstrate that higher levels of expression are achieved using this system than by transfection of plasmids coding for the RBD or Spike. They then report how a single-dose administration of the recombinant virus to influenza-naive mice induces the production of neutralising antibodies against both SARS-CoV-2 and influenza virus.

The manuscript is well written, experiments and results clearly described and the limitations of the study appropriately outlined (lack of evaluation in a more advanced model, use of influenza-naive mice). Overall, the authors present an innovative approach in the design of vaccines targeting the SARS-CoV-2 RBD that would be interesting to be tested in a SARS-CoV-2 challenge study.

Minor comments:

l.144: ... pseudotyped lentivirus with contained... with should be removed
l.172: It would be useful to briefly explain the role of the murine B7.1 in the text.
l.204, Fig 1C, Fig 3C-D: It would be useful to state in the main text, where relevant, in which cell line the virus was passaged (even if it is described in the methods).
l.211: Why was mutation D379W selected over R453G for the final construct since both improved virus titres in vitro?
l.251: It should be Y-axis instead of X-axis.
l.256: A brief description of the lentivirus assay would be useful in the text.

Author Response

We thank the reviewer for their helpful comments, the following revisions have been made to the manuscript with respect to their suggestions:

l.146  “with” has been removed the sentence now reads:

“... pseudotyped lentivirus contained a cytoplasmic…”

l.174-167 . We have added the following statement to the text to further discuss the role of B7.1:

“The murine B7.1 sequence was used as it has previously been shown to induce high levels of surface expression of chimeric proteins [29].” 

l.208-209 We have clarified in the text that the virus was passaged in MDCK/SIAT1/TMPRSS2 cells:

“...was blind passaged twice in MDCK-SIAT1-TMPRSS2 cells at three-day intervals.”

l.216-18 The virus used for murine infections and cell culture surface expression studies was the original isolate, passaged four times, which had accumulated the G379W mutation upon passaging in cells. The R453G mutation was identified in a reverse genetics rescue attempting to incorporate a longer NA segment, this virus was found to have lost the NA segment upon serial passaging. The R453G mutation also improves the ability to grow without NA and may be useful for further genetic engineering of influenza, so we have included it here. However, as this virus grows well without an NA segment, we were concerned that this mutation might alter the stability of the NAflank_RBD segment. As it is important that this segment be maintained, we proceeded with the G379W mutation - which we had evidence had maintained the NAflank_RBD segment over multiple passages. To clarify this a few changes have been made to the text:

l.216 “However, a single amino acid substitution was present in HA (G379W).”

l.218 “...and additional HA mutation isolated in a virus with no NA segment (R453G), improved titers of ΔNA(RBD)-Flu generated by reverse genetics”

l.258 Thank you, this has been fixed. The sentence in line 251 now reads:

“Y-axis values show reciprocal dilution…”

l.259 We have describe the lentivirus neutralization assay in great detail previously (https://doi.org/10.3390/v12050513), we have added a citation to this directly in the main text:

“...CoV-2 titers using a Spike-pseudotyped lentivirus neutralization assay [33,35].”

Reviewer 2 Report

The manuscript by Loes et al discussed the production of antibodies in mice in response to attenuated influenza virions expressing the SARS3 CoV-2 receptor-binding domain. The manuscript is novel and provides useful insights. The authors should clarify the number of replicates in Figure 3 and whether the experiments were repeated for the other figures as well. In addition, what would be the results of using another strain of mice such as B6 mice? Would this lead to any difference in results? Finally, the limitations of the study need to be elaborated upon more directly in the discussion.

Author Response

We thank the reviewer for their comments, in response to their suggestions, we have made a number of changes to the manuscript. 

We have made sure that the number of replicates is stated in the legend for each figure: 

Fig. 1 l.198-199 “Values shown represent titers from a single reverse genetics experiment.” (unchanged)

Fig. 2. l.130-132 “. Histograms shown are for a single well of stained cells. However, verification of surface expression of RBD upon infection with ΔNA(RBD)-Flu was replicated on a separate day.”

l.244 “Data shown is from a single population of stained cells.”

Fig. 3. l.262 “…Blue circles denote titer from individual mice in each group, horizontal line indicates median titer for the group. Samples were run in duplicate.”

We cannot directly comment on how the vaccine would work in another mouse strain. We used BALB/cJ because they are the most commonly used strain in influenza virus infections, which are the type of mouse experiments are lab typically performs. We hypothesize that responses would be similar in B6 mice, but cannot be certain. In the limitations section in the Discussion, we have added text that directly notes that we tested only a single strain of mice.

l.298 “...we performed experiments using only in a single strain of mice (BALB/cJ)”

We have expanded the text in the Discussion (beginning on line 302) on the limitations. The Discussion now notes all the major limitations:

  • We only tested the vaccination in influenza-naive mice, and so cannot comment on interference from pre-existing anti-influenza immunity.
  • We only performed experiments in a single strain of mice, and not more elaborate animal models such as non-human primates.
  • We did not examine the durability of the antibody titers over long time frames
  • We only assessed the formation of neutralizing antibodies, and not if they are protective against challenge with SARS-CoV-2.

Reviewer 3 Report

The paper presented an interesting idea to use attenuated influenza visions to express SARS-CoV-2 RBD domain, which subsequently induced SARS-CoV-2 neutralizing antibodies, as well as anti-influenza neutralizing antibodies in mouse experiments. It is a proof of concept to use influenza virus as a platform to express SARS-CoV-2 protein, which could potentially serve as dual influenza and SARS-CoV-2 vaccines. But there are some limitations to the current design as mostly described in discussion section of the manuscript. There is on more limitation that needs to be paid more attention is the virus titer was really low for the engineered influenza virus with RBD (without NA) as shown in Figure 1C, which was 2-3 orders of magnitude than the virus titer of the influenza virus with NA (without RBD). 

There was no NA in the engineered influenza virus with RBD. So for the virus to be replicated, binding affinity-reduced HA such as Y98F HA mutant had to be used. Interestingly, during the virus passage, two additional HA mutations were found to increase the virus titer: D379W (but this reviewer found G379 in the original HA from A/Aichi/2/1968 (H3N2)) and R453G. No rationale was provided for these two mutations in influenza virus replication, which appear to be in HA2 segment of HA.

The manuscript was clearly written. There was one typo: in page 6, line 215, Fig. 3C should be Fig. 1C.

Author Response

We thank the reviewer for their careful review of the manuscript and have made the several changes to the manuscript in response to their comments.

In reference to the viral titers, this is a good point: the virus without a NA grows to lower titers than virus with NA. We have now mentioned this limitation both when we discuss the titers in the Results and in the limitations section of the Discussion.

l.219-221 “We validated that this mutation, and an additional HA mutation isolated in an NA-deficient virus with no NA segment (R453G), improved titers of ΔNA(RBD)-Flu generated by reverse genetics, although the titers were still lower than for a virus with the same HA and an intact NA segment (Fig 1C).”

l.299-301 “Third, although the ΔNA(RBD)-Flu grew to reasonably high titers that were sufficient for our experiments, the titers were still lower than those obtained using virus with an intact NA segment.”

The reviewer is correct - the wildtype residue at position 379 is a glycine not an asparate, this was an error and has been corrected in the text and figures. Additionally, as this HA mutant may be useful for others working with NA-deficient influenza strains, we have included the genbank file for this plasmid in the supplement. 

We have not done additional experiments to identify how these mutations in the HA2 segment improve growth without NA. A further understanding of how these mutations alter growth of a NA-deficient virus is potentially of interest for further genetic engineering of influenza. However, we believe mechanistic studies regarding the effect of these mutations on viral growth is outside the scope of this paper. 

We have, however, added a statement clarifying that we do not know the mechanism for how these mutations improve rescue in the absence of an NA segment and protein:

l.221-222 “The mechanism for how these mutations in HA2 might improve titers from reverse genetics is not known.”

Finally, we have fixed the typo in line 215 now reads:

l.221 “...improved titers of ΔNA(RBD)-Flu generated by reverse genetics (Fig 1C).”